# Task-agnostic Continual Learning with Hybrid Probabilistic Models

**Polina Kirichenko** [1]  **Mehrdad Farajtabar** [2]  **Dushyant Rao** [2]  **Balaji Lakshminarayanan** [3]  **Nir Levine** [2]  **Ang Li** [2]
**Huiyi Hu** [2]  **Andrew Gordon Wilson** [1]  **Razvan Pascanu** [2]

## Abstract

Learning new tasks continuously without forgetting on a constantly changing data distribution is essential for real-world problems but extremely challenging for modern deep learning. In this work we propose HCL, a **H**ybrid generative-discriminative approach to **C**ontinual **L**earning for classification. We model the distribution of each task and each class with a normalizing flow. The flow is used to learn the data distribution, perform classification, identify task changes, and avoid forgetting, all leveraging the invertibility and exact likelihood which are uniquely enabled by the normalizing flow model. We use the generative capabilities of the flow to avoid catastrophic forgetting through generative replay and a novel functional regularization technique. For task identification, we use state-of-the-art anomaly detection techniques based on measuring the typicality of the model's statistics. We demonstrate the strong performance of HCL on a range of continual learning benchmarks such as split-MNIST, split-CIFAR, and SVHN-MNIST.

## 1. Introduction

For humans, it is natural to learn new skills sequentially without forgetting the skills that were learned previously. Deep learning models, on the other hand, suffer from *catastrophic forgetting*: when presented with a sequence of tasks, deep neural networks can successfully learn the new tasks, but the performance on the old tasks degrades (McCloskey & Cohen, 1989; French, 1999; Kirkpatrick et al., 2017; Parisi et al., 2019; Hadsell et al., 2020). Being able to learn sequentially without forgetting is crucial for numerous applications of deep learning. In real life, data often arrives as a continuous stream, and the data distribution is constantly changing.

For example, consider a neural network that might be used for object detection in self-driving cars. The model should continuously adapt to different environments, e.g. weather and lighting. While the network learns to work under new conditions, it should also avoid forgetting. For example, once it adapts to driving during the winter, it should still work well in other seasons. This example illustrates the *domain-incremental continual learning* setting: the distribution of the inputs to the model evolves over time while the target space stays the same. Moreover, in this scenario, the model should be *task-agnostic*: it has no information on the task boundaries, i.e., the timestamps when the input distribution changes.

Motivated by the task-agnostic domain-incremental continual learning setting, we propose Hybrid Continual Learning (HCL) – an approach based on simultaneous generative and discriminative modeling of the data with normalizing flows. Fig. 1 schematically demonstrates the framework. The contributions of our work are as follows:

- We propose HCL, a normalizing flow-based approach to task-agnostic continual learning.We employ two methods to alleviate catastrophic forgetting: generative replay and a novel functional regularization technique. We provide an empirical comparison and theoretical analysis of the two techniques showing that the functional regularization constrains the model more than generative replay to avoid forgetting, and generally leads to better performance.

- We conduct experiments on a range of image classification continual learning problems on split MNIST, split CIFAR, SVHN-MNIST and MNIST-SVHN datasets. HCL achieves strong performance in all settings.

- We show that HCL can successfully detect task boundaries and identify new as well as recurring tasks, measuring the typicality of model's statistics.

## 2. Background and Notation

**Continual learning (CL)**  We assume that a continual learning model $g_\theta : \mathcal{X} \to \mathcal{Y}$ is trained on a sequence of $\tau$ supervised tasks: $T_{t_1}, T_{t_2}, \ldots, T_{t_\tau}$. Each task

[1]New York University [2]DeepMind [3]Google Brain. Correspondence to: Polina Kirichenko <pk1822@nyu.edu>, Mehrdad Farajtabar <farajtabar@google.com>.

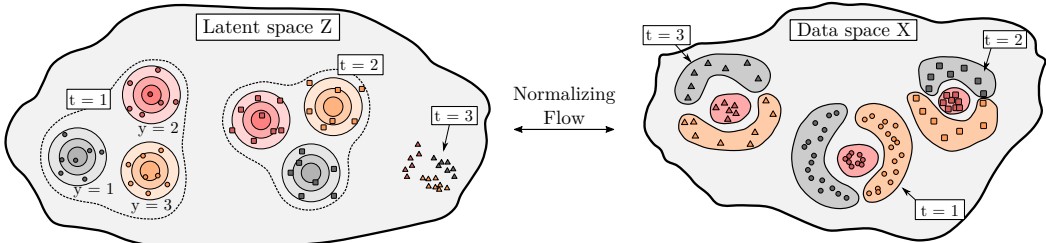

*Figure 1.* An illustration of the proposed Hybrid Continual Learning (HCL) framework. HCL models the distribution of each class in each task as a latent Gaussian distribution transformed by a normalizing flow. We show the Gaussian mixtures corresponding to the two tasks $t_1$ and $t_2$ in the latent space on the left, and the corresponding data distributions on the right. If a new task $t = 3$ appears, HCL identifies it using the typicality of the flow's statistics, and initializes the Gaussian mixture for a new task.

$T_i = \{(x_j^i, y_j^i)\}_{j=1}^{N_i}$ has the input space $\mathcal{X}^i$, the label space $\mathcal{Y}^i$, and the corresponding data-generating distribution $p_i(x, y)$. The number of tasks $\tau$ is not known in advance, and while training on a task $T_i$ the model does not have access to the data from previous $T_1, \ldots, T_{i-1}$ or future tasks $T_{i+1}, \ldots, T_\tau$. The objective of a CL model is to minimize $\sum_{i=1}^M E_{x,y \sim p_i(\cdot,\cdot)} l(g_\theta(x), y)$ for some risk function $l(\cdot, \cdot)$, and thus, generalize well on all tasks after training. In this work, we focus on classification, and in particular, the domain-incremental learning setting with $\mathcal{Y}^i = \{1, \ldots K\}$ for all tasks $i$. For more on CL settings see (van de Ven & Tolias, 2019) and (Hsu et al., 2018).

**Task-agnostic CL** In most continual learning algorithms, it is crucial to know the task boundaries — the moments when the training task is changed. At each iteration $j$ of training, we receive a tuple $(x(j), y(j), t(j))$ where $x(j)$ and $y(j)$ is a batch of data and the corresponding labels and $t(j)$ is the index of the current task. In this work, we also consider the *task-agnostic* setting, where the task index $t(j)$ is not provided and the algorithm has to infer it from data.

## 3. Hybrid Model for Continual Learning

### 3.1. Modeling the data distribution

HCL approximates the data distribution with a single normalizing flow, with each class-task pair $(y, t)$ corresponding to a unique Gaussian in the latent space (see Fig. 1 for illustration). More precisely, we model the joint distribution $p_t(x, y)$ of the data $x$ and the class label $y$ conditioned on a task $t$ as $p_t(x, y) \approx \hat{p}(x, y|t) = \hat{p}_X(x|y, t)\hat{p}(y|t)$, where $\hat{p}_X(x|y, t)$ is modeled by a normalizing flow $f_\theta$ with a base distribution $\hat{p}_Z = \mathcal{N}(\mu_{y,t}, I) : \hat{p}_X(x|y, t) = f_\theta^{-1}(\mathcal{N}(\mu_{y,t}, I))$. Here $\mu_{y,t}$ is the mean of the latent distribution corresponding to the class $y$ and task $t$. We assume that $\hat{p}(y|t)$ is a uniform distribution over the classes for each task: $\hat{p}(y|t) = \frac{1}{K}$.

We train the model by maximum likelihood: for each minibatch of data $(x(j), y(j), t(j))$ we compute the likelihood using the change of variable formula and take a gradient

step with respect to the parameters $\theta$ of the flow. In the task-agnostic setting, we have no access to the task index $t_j$ and instead infer it from data (see Section 3.2). At test-time, HCL classifies an input $x$ to the class $\hat{y}$ using the Bayes rule: $\hat{p}(y|x) \propto \hat{p}(x|y)$, so $\hat{y} = \arg\max_y \sum_{t=1}^\tau \hat{p}_X(x|y, t)$. Notice that we do not have access to the task index at test time, so we marginalize the predictions over all tasks $t$.

### 3.2. Task Identification

In the task-agnostic scenario, the task identity $t$ is not given during training and has to be inferred from the data. The model starts with $K$ Gaussians with means $\{\mu_{y,t_1}\}_{y=1}^K$ in the latent space corresponding to the classes of the first task. We assume that a model first observes batches of data $B_1, \ldots, B_m$ from the task $T_{t_1}$ where each $B = \{(x_j, y_j)\}_{j=1}^b$. Then, at some unknown point in time $m + 1$, it starts observing data batches $B_{m+1}, B_{m+2}, \ldots$ coming from the next task $T_{t_2}$. The model has to detect the task boundary and initialize Gaussian mixture components in the latent space which will correspond to this new task $\{\mathcal{N}(\mu_{y,t_2}, I)\}_{y=1}^K$. Moreover, in our set-up some of the tasks can be recurring. Thus, after observing tasks $T_{t_1}, \ldots, T_{t_k}$ and detecting the change point from the task $T_k$, the model has to identify whether this batch of data comes from a completely new task $T_{t_{k+1}}$ (and add new Gaussians for this task in the latent space) or from one of the previous tasks $T_{t_1}, \ldots, T_{t_{k-1}}$.

Similarly to prior work on anomaly detection (Nalisnick et al., 2019c) and (Morningstar et al., 2020), we detect task changes measuring the typicality of the HCL model's statistics. Following Morningstar et al. (2020), we can use the following statistics on data batches $B$: log-likelihood $S_1(B, t) = \sum_{(x_j, y_j) \in B} \hat{p}_X(x_j|y_j, t)$, log-likelihood of the latent variable $S_2(B, t) = \sum_{(x_j, y_j) \in B} \hat{p}_Z(f(x_j)|y_j, t)$ and log-determinant of the Jacobian $S_3(B, t) = S_1(B, t) - S_2(B, t)$. For each task $t$, we keep track of the mean $\mu_S^t$ and the standard deviation $\sigma_S^t$ for these statistics over a window of the last $l$ batches of data. Then, if any statistic $S(B, t)$ of the current batch $B$ and task $t$ is not falling within the typical

set $|S(B, t) - \mu_S^t| > \lambda\sigma_S^t$, HCL detects a task change. In this case, if all the statistics are in the typical set $|S(B, t') - \mu_S| < \lambda\sigma_S$ for one of the previous tasks, we identify a switch to the task $t'$; otherwise, we switch to a new task. In practice, for most standard CL benchmarks such as split-MNIST we only use a single statistic – HCL's log-likelihood which is sufficient for robust task change detection. However, for more challenging scenarios identified in Nalisnick et al. (2019a), we use all three statistics described above.

### 3.3. Alleviating Catastrophic Forgetting

#### 3.3.1. GENERATIVE REPLAY

Following Shin et al. (2017); Rao et al. (2019), we train the model on the mix of real data from the current task and generated data from previous tasks to combat forgetting. For generating the replay data, we store a single snapshot of the HCL model $\hat{p}_X^{(k)}(x|y, t)$ with weights $\theta^{(k)}$ taken at a point of the last detected task change $T_{t_k} \to T_{t_{k+1}}$. We generate and replay data from old tasks using the snapshot: $x_{GR} \sim \hat{p}_X^{(k)}(x|y, t)$, where $y \sim U\{1, \dots, K\}$ and $t \sim U\{t_1, \dots, t_k\}$, and maximize its likelihood $\mathcal{L}_{GR} = \log \hat{p}_X(x_{GR}|y, t)$ under the current HCL model $\hat{p}_X(\cdot)$. We store only a *single* snapshot model throughout training as it approximates the data distribution of all tasks up to $T_{t_k}$. After detecting the task change $T_{t_{k+1}} \to T_{t_{k+2}}$, we update the snapshot with new weights $\theta^{(k+1)}$. The resulting objective function in generative replay training is $\mathcal{L}_{ll} + \mathcal{L}_{GR}$, where $\mathcal{L}_{ll}$ is the log-likelihood of the data on the current task. See Appendix D for a further discussion of the generative replay objective. We refer to HCL with generative replay as HCL-GR. In prior work, generative replay has been successfully applied, predominantly using GANs or VAEs (Shin et al., 2017; Rao et al., 2019; Lee et al., 2020; Ye & Bors, 2020; Pomponi et al., 2020b; Mundt et al., 2019; Achille et al., 2018).

#### 3.3.2. FUNCTIONAL REGULARIZATION

We propose a novel functional regularization loss that enforces the flow to map samples from previous tasks to the same latent representations as a snapshot model. Specifically, similar to GR, we save a snapshot of the model $\hat{p}_X^{(k)}(\cdot)$ taken after detecting a shift from the task $T_{t_k}$ and produce samples $x_{FR} \sim \hat{p}_X^{(k)}(x|y, t)$ for $y \sim U\{1, \dots, K\}$, $t \sim U\{t_1, \dots, t_k\}$. However, instead of generative replay loss $\mathcal{L}_{GR}$, we add the following term to the maximum likelihood objective $\mathcal{L}_{FR} = \|f_\theta(x_{FR}) - f_{\theta^{(k)}}(x_{FR})\|^2$, where $f_\theta$ is the current flow mapping and $f_{\theta^{(K)}}$ is the snapshot model. We note that the $L_2$ distance in $\mathcal{L}_{FR}$ is a natural choice given the choice of $p_Z(z|y, t)$ as a Gaussian, as $\mathcal{L}_{ll}$ also contains a linear combination of losses of the form $\|f_\theta(x) - \mu_{y,t}\|^2$. The term $\mathcal{L}_{FR}$ can be understood as con-

trolling the amount of change we allow for the function $f$, hence controlling the trade-off between stability and plasticity. In practice, we weigh the term by $\alpha$: $\mathcal{L}_{ll} + \alpha\mathcal{L}_{FR}$. We refer to the method as HCL-FR. To the best of our knowledge, the loss $\mathcal{L}_{FR}$ is novel: it is designed specifically for normalizing flows and cannot be trivially extended to other generative models. In order to apply $\mathcal{L}_{FR}$ to VAEs, we would need to apply the loss separately to the encoder and the decoder of the model, and potentially to their composition. Recently, Titsias et al. (2019) and Pan et al. (2020) proposed related regularization techniques for continual learning which rely on the Gaussian Process framework.

**Theoretical analysis** In Appendix E.1 we study the loss $\mathcal{L}_{FR}$ theoretically and draw connections to other objectives. In particular, the term can be interpreted as looking at the amount of change in the function as measured by the KL-divergence assuming the output of the flow is an isotropic Gaussian. Under a Taylor approximation, we show that $\mathcal{L}_{FR}$ enforces the weights to move only in directions of low curvature of the mapping $f_\theta^{(k)}$ when learning a new task. Hence, similar to regularization-based CL methods, this term limits movement of the weights in directions that lead to large functional changes of the flow.

## 4. Experiments

In this section, we evaluate HCL on a range of image classification tasks in continual learning. In all experiments, we consider domain-incremental learning where the number of classes $K$ is the same in all tasks. At test time, the task identity is *not* provided to any of the considered methods. For HCL, we report the performance both in the task-aware (when the task identity is provided to the model during training) and task-agnostic (no task boundary knowledge during training) settings. We use RealNVP and Glow normalizing flow models. See Appendix A for detailed setup.

**Metrics** Let $a_{i,j}$ be the accuracy of the model on task $i$ after training on $j$ tasks. We report the following metrics: (1) final accuracy $a_{i,\tau}$ on each task $i \in \{1, \dots \tau\}$ at the end of the training, (2) average final accuracy across tasks $\frac{1}{d}\sum_{i=1}^{d} a_{i,\tau}$, (3) the average forgetting: $\frac{1}{\tau-1}\sum_{i=1}^{\tau-1}(a_{i,i} - a_{i,\tau})$, and (4) the overall accuracy: the final accuracy on $(K \times \tau)$-way classification which indicates how well the model identifies both the class and the task. We run each experiment with 3 seeds and report mean and standard deviation of the metrics.

**Adam** We evaluate Adam training without any extra steps for preventing catastrophic forgetting.

**Multi-Task Learning (MTL)** We evaluate multitask learning (MTL): the model is trained on each task $T_{t_i}$ for the same number of epochs as in CL methods, however, when training on $T_{t_i}$, it has access to all previous tasks

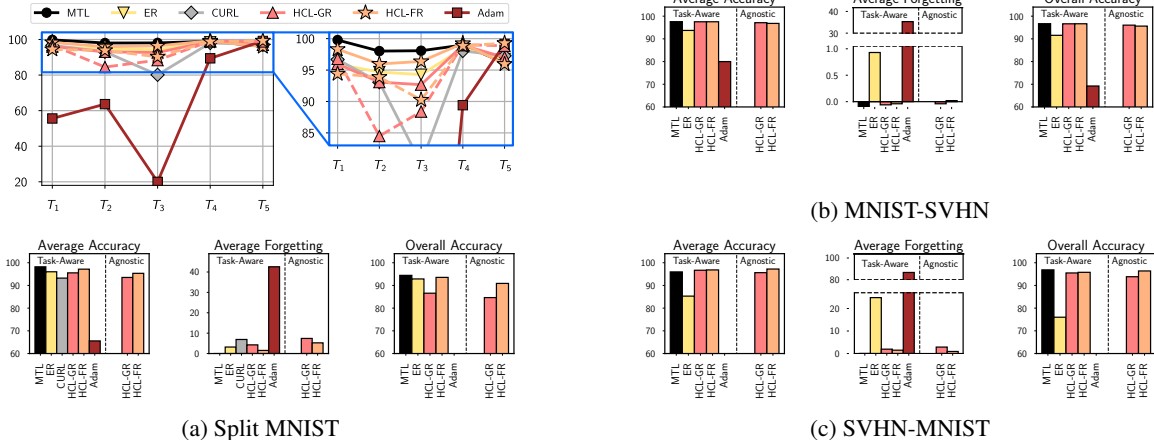

(a) Split MNIST          (b) MNIST-SVHN          (c) SVHN-MNIST

*Figure 2.* Results on **(a)** Split MNIST, **(b)** MNIST-SVHN and **(c)** SVHN-MNIST image datasets. For Split MNIST, in the top panel we show the performance of each method on each of the tasks in the end of training; for HCL we show the results in the task-agnostic setting with dashed lines. We also show average accuracy, forgetting and overall accuracy for each of the datasets and methods. HCL provides strong performance, especially on SVHN-MNIST where it achieves almost zero forgetting and significantly outperforms ER.

$T_{t_1}, \ldots T_{t_{i-1}}$. At each iteration, we sample a mini-batch (of the same size as the current data batch) containing the data from each of the tasks that have been observed so far.

**Experience Replay (ER)**    We reserve a buffer with a fixed size of 1000 samples for each task and randomly select samples to add to that buffer during training on each task. When training on task $T_k$, the model randomly picks a number of samples equal to the current task's batch size from each of the previous task buffers and appends to the current batch.

**CURL**    We evaluate the state-of-the-art CURL (Rao et al., 2019) method for continual learning which is most closely related to HCL: CURL also incorporates a generative model (VAE), with an expanding Gaussian mixture in latent space, and likelihood-based task-change detection.

**Split MNIST**    In this experiment, following prior work we split the MNIST dataset (LeCun et al., 1998) into 5 binary classification tasks. We train for 30 epochs on each task. We use the Glow architecture to model the data distribution. The results are presented in Fig 2 (a) and Appendix Table 1. HCL shows strong performance, competitive with ER. Out of the HCL variants, HCL-FR provides a better performance both in the task-aware and the task-agnostic settings. Both HCL variants significantly outperform CURL. We hypothesise that since it only uses a single latent Gaussian component for each class, CURL cannot as easily capture a highly multimodal and complex data distribution for a single class – a requirement for domain-incremental learning where classes may be visually very different across different tasks. In contrast, as HCL initialises multiple latent components in a task-agnostic fashion and draws upon a flexible flow-based model, it is much better suited to the domain-incremental continual learning setting. The final accuracy of the Adam baseline on some tasks is very low: unless we take measures

to avoid forgetting, the flow typically maps the data from all tasks to the region in the latent space corresponding to the final task, and it may happen that e.g. the data in class 1 of the first task will be mapped to the class 2 of the last task.

**MNIST-SVHN and SVHN-MNIST**    We evaluate HCL and the baselines on two more challenging problems: MNIST-SVHN and SVHN-MNIST. Here, the tasks are 10-way classification problems on either the SVHN (Netzer et al., 2011) or the MNIST dataset. We use the RealNVP architecture with inputs of size $32 \times 32 \times 3$, and upscale the MNIST images to this resolution. We train the methods for 90 epochs on each task. We report the results in Fig 2 (b) and (c) and Appendix Table 2. HCL-FR and HCL-GR show strong performance, outperforming ER and Adam significantly, and performing on par with MTL. On MNIST-SVHN, the model is able to almost completely avoid forgetting.

See Appendix B for experimental results on split CIFAR-10 and split CIFAR-100 and Appendix G discussing task identification results.

## 5. Discussion

In this work we proposed HCL, a hybrid model for continual learning based on normalizing flows. HCL achieves strong performance on a range of image classification problems and is able to automatically detect new and recurring tasks using the typicality of flow's statistics. We believe that the key advantage of HCL is its simplicity and extensibility. HCL describes the data generating process using a tractable but flexible probabilistic model and uses the maximum-likelihood to train the model.

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

# Task-agnostic Continual Learning with Hybrid Probabilistic Models: Supplementary Material

## A. Setup and hyperparameters

### A.1. HCL

**Setup**   We use RealNVP and Glow normalizing flow models. We initialize the class-conditional latent Gaussian means $\mu_{y,t}$ randomly when a new task is identified. Following Izmailov et al. (2020) we do not train $\mu_{y,t}$ and instead keep them fixed. We use the Adam optimizer (Kingma & Ba, 2014) to train the parameters of the flow and a batch size of 32. For both GR and FR, we generate the number of replay samples equal to the batch size for each of the previously observed tasks. For task identification, we use a window of $l = 100$ mini-batches and the sensitivity $\lambda$ in threshold for task detection is set to 5. For more details on the hyper-parameters, please see the Appendix A.

**Split MNIST**   For split MNIST, we use a Glow architecture (Kingma & Dhariwal, 2018), generally following Nalisnick et al. (2019a) for the model setup. We use Highway networks (Srivastava et al., 2015) as coupling layer networks which predicted the scale and shift of the affine transformation. Each Highway network had 3 hidden layers with 200 channels. We use the Glow with multi-scale architecture with 2 scales and 14 coupling layers per scale, squeezing the spatial dimension in between two scales.

For the sensitivity parameter $\lambda$, we tested values $\lambda = 3$ and $\lambda = 5$ on split MNIST dataset. For the lower value of $\lambda$ the model correctly identified the actual task shifts, however, it detected a higher number of extra task shifts. We used $\lambda = 5$ for the rest of the experiments.

**SVHN-MNIST and MNIST-SVHN**   We use a RealNVP (Dinh et al., 2017) model with ResNet-like coupling layer network for SVHN-MNIST and MNIST-SVHN experiments. The ResNet networks have 8 blocks with 64 channels, and use Layer Normalization (Ba et al., 2016) instead of Batch Normalization (Ioffe & Szegedy, 2015). RealNVP has 3 scales and 16 coupling layers in total.

**CIFAR embeddings**   For this set of experiments discussed in Appendix B, we use RealNVP model with 1 scale with 8 coupling layers, and MLP coupling layer networks which has 3 hidden layers and 512 hidden units in each layer.

We use the weight decay $5 \times 10^{-5}$ on split MNIST, SVHN-MNIST and MNIST-SVHN experiments, and tune the weight decay om the range $\{10^{-4}, 10^{-3}, 10^{-2}\}$ on a validation set.

For HCL-FR on split MNIST, SVHN-MNIST and MNIST-SVHN we set the weight of the regularization term of $\mathcal{L}_{FR}$ objective $\alpha = 1$. For split CIFAR-10 and split CIFAR-100, we tune $\alpha$ on a validation set in the range $\{1, 5, 10, 100\}$. Generally, we do not notice major difference in performance of HCL-FR and its task-agnostic version when varying $\alpha$.

We compare our HCL-GR and HCL-FR to other training procedures for the same flow model: regular Adam training, multi-task learning and experience replay. Additionally, we compare to CURL (Rao et al., 2019) which is based on a VAE architecture.

**Adam**   As a baseline, we train HCL model with Adam optimizer. Prior work (Mirzadeh et al., 2020; Hsu et al., 2018) argues against using Adam for continual learning. However, it is challenging to train normalizing flows with SGD. We experimented with Adagrad (Duchi et al., 2011) and RMSProp but did not observe a significant improvement compared to Adam.

**Experience Replay**   Note that we fix the size of the buffer per task throughout the experiments, resulting in varying performance: on the SVHN-MNIST the total size of the buffer is only about $1.5\%$ of the SVHN dataset, while on Split CIFAR-100 by the end of training the size of the combined buffer on all tasks is $20\%$ of the dataset.

### A.2. CURL

We evaluate the state-of-the-art CURL (Rao et al., 2019) method for continual learning which is most closely related to HCL: CURL also incorporates a generative model (VAE), with an expanding Gaussian mixture in latent space, and likelihood-based task-change detection. However, in the original paper the method is only evaluated in class- and task-incremental learning settings, and focuses on unsupervised learning. To provide a fair comparison, we use the *supervised* variant of CURL proposed by Rao et al. (2019), in which the label is used to directly train the corresponding Gaussian component in latent

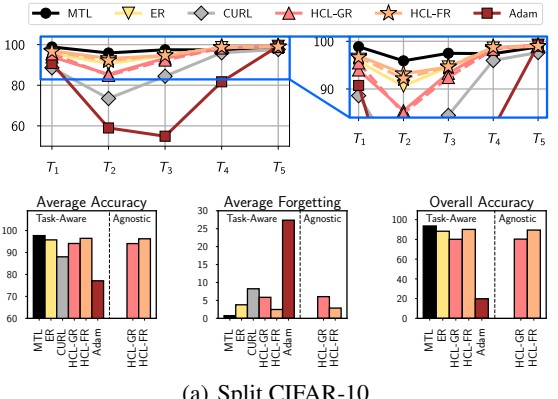

(a) Split CIFAR-10

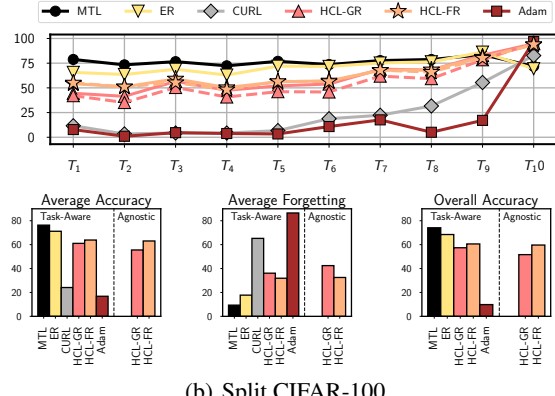

(b) Split CIFAR-100

*Figure 3.* Results on Split CIFAR embedding datasets. We use embeddings extracted by an EfficientNet model (Tan & Le, 2019) pre-trained on ImageNet. In the top panels we show the performance of each method on each of the tasks in the end of training; for HCL we show the results in the task-agnostic setting with dashed lines. At the bottom, we show average accuracy, forgetting and overall accuracy for each of the methods. HCL outperforms CURL and Adam and performs on par with experience replay with a large replay buffer. HCL-FR provides better performance than HCL-GR.

space. It is important to note that while CURL is task-agnostic in the unsupervised setting, it implicitly infers the task via labels in the supervised task- and class-incremental settings (and hence does not perform task-change detection or unsupervised expansion). Thus, for domain-incremental learning (where the labels do not signal the introduction of a new task), we snapshot the generative model on task change, meaning that it is not task-agnostic in this setting. Finally, since the supervised variant of CURL utilises a single Gaussian component in latent space for each label, we cannot innately compute the overall accuracy with respect to class labels after training on domain labels.

We train all models with the Adam optimizer, with a learning rate of $10^{-3}$. For MNIST, we use the same architecture as in Rao et al. (2019), with a MLP encoder with layer sizes $[1200, 600, 300, 150]$, a MLP Bernoulli decoder with layer sizes $[500, 500]$, and a latent space dimensionality of 32. For CIFAR10/100 features, we use 2-layer MLPs for both the encoder and decoder (with a Gaussian likelihood for the decoder), with 512 units in each layer and a latent space dimensionality of 64.

# B. Additional experimental results

**Split CIFAR embeddings**     We consider the Split CIFAR-10 and Split CIFAR-100 datasets, where each task corresponds to 2 classes of CIFAR-10 and 10 classes of CIFAR-100 (Krizhevsky et al., 2009) respectively. Generative models typically struggle to generate high fidelity images when trained on CIFAR datasets due to their high variance and low resolution. For this reason, we utilize transfer learning and, instead of using the raw pixel values, use embeddings extracted by an EfficientNet model (Tan & Le, 2019) pre-trained on ImageNet. Normalizing flows have been shown to perform well on classification and out-of-distribution detection tasks using the image embeddings (Izmailov et al., 2020; Kirichenko et al., 2020; Zhang et al., 2020; Zisselman & Tamar, 2020). We report the results in Fig. 3 and Appendix Tables 3 and 4. We train all methods for 15 epochs per task. In the Appendix Tables, we additionally report the performance in the single-pass setting training for just one epoch per task. HCL provides strong performance, with functional regularization giving the best results. ER provides a strong baseline, especially on CIFAR-100 due to the relatively large (20% of the dataset) size of the replay buffer. CURL underperforms compared to the HCL variants. Since CURL is a VAE-based model, it requires an appropriate decoder likelihood for each data type (a Gaussian distribution in the case of CIFAR embeddings), while our flow-based HCL model directly models the distribution in the data space without extra approximations. We hypothesize that this advantage is one of reasons for the HCL superior performance.

In Tables 1, 2, 3 and 4, we provide detailed results of the experiments. For each experiment and method with the exception of MNIST-SVHN and SVHN-MNIST we repeat the experiments three times with different random initializations. We report the mean and standard deviation of the results. We report the results on Split MNIST in Table 1; on SVHN-MNIST and MNIST-SVHN in Table 2; on Split CIFAR-10 in Table 3 and on Split CIFAR-100 in Table 4. For Split CIFAR datasets we report the results in two settings: with 15 epochs per task and with 1 epoch per task (single-pass).

*Table 1.* Results of the experiments on **split MNIST** dataset with MTL (multitask learning), Adam (regular training without alleviating forgetting), ER (standard experience or data buffer replay with the capacity of 1000 samples per task), HCL-GR (generative replay), HCL-FR (functional regularization). The dataset with 10 classes is split into 5 binary classification tasks, as well as task-agnostic versions of HCL-FR and HCL-GR.

| Task # | 1 | 2 | 3 | 4 | 5 | Acc Avg | Forget Avg | Full Acc |
|---|---|---|---|---|---|---|---|---|
| MTL | 99.78 $\pm0.15$ | 98.02 $\pm1.50$ | 98.08 $\pm0.96$ | 98.98 $\pm0.33$ | 96.15 $\pm1.87$ | 98.20 $\pm0.88$ | $-1.32$ $\pm1.03$ | 94.44 $\pm1.06$ |
| Adam | 55.59 $\pm4.74$ | 63.66 $\pm3.10$ | 19.96 $\pm7.03$ | 89.41 $\pm7.15$ | 99.16 $\pm0.41$ | 65.56 $\pm1.33$ | 42.57 $\pm1.49$ | 19.66 $\pm0.08$ |
| ER | 95.92 $\pm5.00$ | 94.69 $\pm1.98$ | 94.27 $\pm2.20$ | 98.44 $\pm0.41$ | 96.77 $\pm1.00$ | 96.02 $\pm1.31$ | 3.19 $\pm1.92$ | 92.86 $\pm1.92$ |
| CURL | 96.67 $\pm0.64$ | 93.06 $\pm1.42$ | 80.14 $\pm5.70$ | 98.05 $\pm0.86$ | 97.27 $\pm0.41$ | 93.23 $\pm1.06$ | 6.90 $\pm1.47$ | – |
| HCL-FR | 98.31 $\pm1.03$ | 95.97 $\pm0.81$ | 96.37 $\pm1.06$ | 99.24 $\pm0.08$ | 95.95 $\pm3.04$ | 97.17 $\pm0.65$ | 1.53 $\pm0.91$ | 93.55 $\pm1.40$ |
| HCL-GR | 95.97 $\pm3.65$ | 93.08 $\pm4.17$ | 92.67 $\pm2.43$ | 98.94 $\pm0.66$ | 97.02 $\pm1.69$ | 95.54 $\pm1.21$ | 4.25 $\pm1.99$ | 86.58 $\pm7.37$ |
| HCL-FR (TA) | 94.41 $\pm3.42$ | 93.88 $\pm0.37$ | 90.25 $\pm4.11$ | 98.86 $\pm0.23$ | 99.28 $\pm0.23$ | 95.33 $\pm0.67$ | 5.24 $\pm0.95$ | 90.89 $\pm0.96$ |
| HCL-GR (TA) | 96.78 $\pm0.98$ | 84.49 $\pm5.57$ | 88.38 $\pm4.79$ | 99.04 $\pm0.53$ | 98.87 $\pm0.06$ | 93.52 $\pm1.84$ | 7.41 $\pm2.18$ | 84.65 $\pm3.46$ |

*Table 2.* Results of the experiments on **SVHN-MNIST** and **MNIST-SVHN** datasets with MTL (multitask learning), Adam (regular training without alleviating forgetting), ER (standard experience or data buffer replay with the capacity of 1000 samples per task), HCL-GR (generative replay), HCL-FR (functional regularization), as well as task-agnostic versions of HCL-FR and HCL-GR.. Each dataset contains two 10-way classification tasks corresponding to MNIST and SVHN.

| | | | SVHN-MIST | | | | | | MNIST-SVHN | | |
|---|---|---|---|---|---|---|---|---|---|---|---|
| Task # | 1 | 2 | Acc Avg | Forget Avg | Full Acc | Task # | 1 | 2 | Acc Avg | Forget Avg | Full Acc |
| MTL | 95.96 | 99.18 | 97.57 | 0.01 | 96.86 | MTL | 99.58 | 95.56 | 97.57 | -0.11 | 96.68 |
| Adam | 9.28 | 99.18 | 54.23 | 86.69 | 30.74 | Adam | 64.16 | 95.82 | 79.99 | 35.31 | 69.23 |
| ER | 71.14 | 99.45 | 85.295 | 24.83 | 76.00 | ER | 98.54 | 88.89 | 93.72 | 0.93 | 91.56 |
| HCL-FR | 94.48 | 99.32 | 96.90 | 1.49 | 95.78 | HCL-FR | 99.51 | 95.55 | 97.53 | -0.04 | 96.65 |
| HCL-GR | 94.03 | 99.35 | 96.69 | 1.94 | 95.5 | HCL-GR | 99.53 | 95.52 | 97.53 | -0.06 | 96.63 |
| HCL-FR (TA) | 95.19 | 99.26 | 97.23 | 0.92 | 96.38 | HCL-FR (TA) | 99.47 | 94.14 | 96.81 | 0.02 | 95.62 |
| HCL-GR (TA) | 91.76 | 99.50 | 95.63 | 2.87 | 93.84 | HCL-GR (TA) | 99.56 | 94.68 | 97.12 | -0.04 | 96.04 |

## C. Differences between HCL-GR and HCL-FR

Intuitively, HCL-FR imposes a stronger restriction on the model. Indeed, in order to have low values of the objective $\mathcal{L}_{FR}$, the model $f_\theta$ has to map the replay data to the same locations in the latent space as the snapshot model $f_\theta^{(k)}$. On the other hand, to achieve the low value of the HCL-GR objective we only need the likelihood of the replay data to be high according to $f_\theta$. In other words, HCL-GR only restricts the locations of $f_\theta(x)$ (for replayed data) to be in the high-density set of $\hat{p}_Z$, but not in any particular position (see Figure 4(a)).

We visualize the effect of both objectives on a two-dimensional *two moons* dataset in Figure 4. We treat the two moons as different tasks and train the HCL model on the top moon shown in grey first. Then, we continue training on the second moon shown in orange using either FR or GR to avoid forgetting. To build an understanding of the effect of these methods, we only use a fixed set of four data points shown as coral squares as the replay data. We show the learned distributions after training on the second task for HCL-GR and HCL-FR in panels (c) and (d) of Figure 4. With a limited number of replay samples, HCL-GR struggles to avoid forgetting. The method is motivated to maximize the likelihood of the replay data, and it overfits to the small replay buffer, forgetting the structure of the first task. HCL-FR on the other hand preserves the structure of the first task better using the same 4 replay samples. In the panel (e) of Figure 4 we visualize the positions to which the models map the replay data in the latent space. HCL-FR (shown with stars) maps the replay data to exactly the

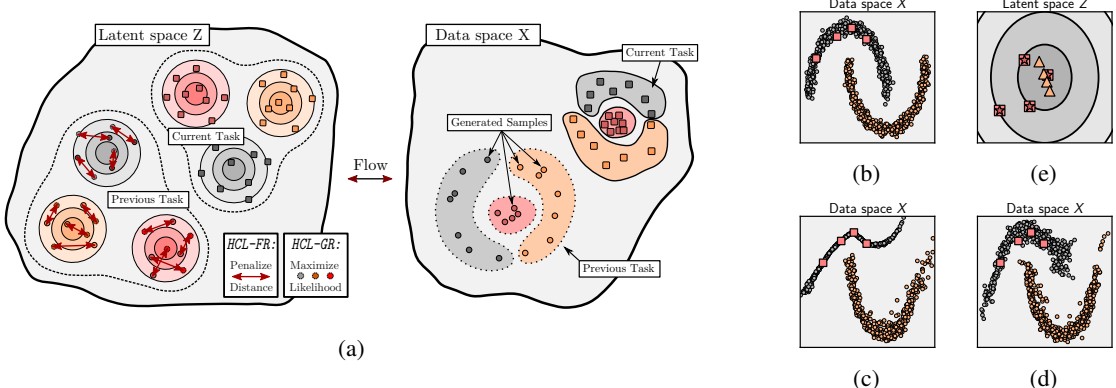

*Figure 4.* Comparison of functional regularization and generative replay. **(a)**: A visualization of HCL-FR and HCL-GR; HCL-GR forces the model to maintain high likelihood of the replay data, while HCL-FR penalizes the distance between the locations of the latent representations for the sampled data for the current and snapshot models. **(b)**: Two moons dataset; data from the first and second tasks is shown with grey and orange circles, and coral squares show the replay samples. **(c)**: Learned distribution after training on the second task with HCL-GR, and **(d)** HCL-FR. **(e)**: Locations of images of the replay data in the latent space for the model trained on the first task (**squares**), HCL-GR (**triangles**) and HCL-FR (**stars**). HCL-FR restricts the model more than GR: the locations of replay samples in the latent space coincide for HCL-FR and the model trained on the first task. Consequently, HCL-FR preserves more information about the structure of the first task.

same locations as the snapshot model $f_\theta^{(1)}$ trained on the first task (shown with squares). HCL-GR (shown with triangles) simply maps the samples to the high-density region without preserving their locations.

To sum up, HCL-FR provides a stronger regularization than HCL-GR while preserving the flexibility of the model, which is crucial for avoiding forgetting. We provide an empirical comparison of HCL-FR and HCL-GR in Section 4 where HCL-FR shows better performance across the board. We discuss the relation between HCL-FR and HCL-GR objectives further in Appendix E.2.

## D. Alleviating forgetting

The loss $\mathcal{L}_{GR}$ in HCL-GR is computed as follows:

$$x_{GR} \sim \hat{p}_X^{(k)}(x|y,t), \quad y \sim U[1,\dots K], \ t \sim U\{t_1,\dots,t_k\}$$

$$\mathcal{L}_{GR} = \log p_X(x_{GR}|y,t) = \log p_Z\left(f_\theta(x_{GR})|y,t\right) + \log\left|\frac{\partial f_\theta}{\partial x}\right|$$

$$= -\frac{1}{2}\|f_\theta(x_{GR}) - \mu_{y,t}\|^2 + \log\left|\frac{\partial f_\theta}{\partial x}\right| + \text{const}. \tag{1}$$

We note that up to a constant, the loss in Eq. (1) can be expressed as the $KL$-divergence between the distribution $\hat{p}^{(k)}$ corresponding to the snapshot model $f_{\theta^{(k)}}$ and the distribution $\hat{p}$ corresponding to the corrent model $f_\theta$:

$$KL[\hat{p}^{(k)}||\hat{p}] = \int \hat{p}^{(k)}(x|y,t) \log \frac{\hat{p}^{(k)}(x|y,t)}{\hat{p}(x|y,t)} dx \approx \frac{1}{J}\sum_{i=1}^{J} \log \frac{\hat{p}^{(k)}(x_i|y,t)}{\hat{p}(x_i|y,t)}$$

$$= \frac{1}{J}\sum_{i=1}^{J}\left(\log \hat{p}^{(k)}(x_i|y,t) \underbrace{- \log \hat{p}(x_i|y,t)}_{GR}\right). \tag{2}$$

## E. Analysis of Functional regularization

In this section, we look at the interpretation of the functional regularization and its connection to regularization based CL methods as well as with generative replay.

### E.1. Taylor expansion of the functional regularization

For simplicity let the flow model be given by

$$f : \mathcal{R}^{M \times N} \to \mathcal{R}^N,$$

where $\mathcal{R}^M$ is the parameter space, and $\mathcal{R}^N$ is the input space. Specifically we have $f(\theta, x) = z$ for the parameters $\theta$ and input $x$. And by abuse of notation let $f^{-1}$ be the inverse flow, namely

$$f^{-1}(\theta, f(\theta, x)) := x.$$

Note that the only thing we did is to make the dependency of the flow on its parameters $\theta$ explicit.

The regularization term that we rely on has the following form

$$\mathcal{L}_{FR} = \frac{1}{J} \sum_{z_j} \left[ \left( z_j - f(\theta_{k+1}, f^{-1}(\theta_k, z_j)) \right)^2 \right], \tag{3}$$

where $\theta_{k+1}$ and $\theta_k$ are the parameters of the model after and before learning the $k$-th task.

For legibility, let $x_j = f^{-1}(\theta_k, z_j)$ and $\theta_{k+1} = \theta_k + \Delta$. We can re-write the loss as $(z_j - f(\theta_{k+1}, x_j))^2$, for a given $z_j$. We will also drop the subscript $j$ when not needed.

We start by taking a first order Taylor expansion around $\theta_k$ of $f(\theta_{k+1}, x)$:

$$\begin{aligned} f(\theta_{k+1}, x) &= f(\theta_k + \Delta, x) \\ &\approx f(\theta_k, x) + \Delta^T \nabla f|_{\theta_k, x}. \end{aligned} \tag{4}$$

We can now re-write the regularizer as:

$$\begin{aligned} \mathcal{L}_{FR} &= \frac{1}{J} \sum_{z_j} \left( \left( z_j - f(\theta_k, x_j) - \Delta^T \nabla f|_{\theta_k, x_j} \right)^2 \right) \\ &= \frac{1}{J} \sum_{z_j} \left[ \left( z_j - f(\theta_k, f^{-1}(\theta_k, z_j)) - \Delta^T \nabla f|_{\theta_k, x_j} \right)^2 \right] \\ &= \frac{1}{J} \sum_{x_j} \left[ \left( \Delta^T \nabla f|_{\theta_k, x_j} \right)^2 \right] \\ &= \frac{1}{J} \sum_{x_j} \left[ \Delta^T \nabla f|_{\theta_k, x_j} (\nabla f|_{\theta_k, x_j})^T \Delta \right] \\ &= \frac{1}{J} \Delta^T \sum_{x_j} \left[ \nabla f|_{\theta_k, x_j} (\nabla f|_{\theta_k, x_j})^T \right] \Delta \\ &= \frac{1}{J} (\theta_{k+1} - \theta_k)^T \sum_{x_j} \left[ \nabla f|_{\theta_k, x_j} (\nabla f|_{\theta_k, x_j})^T \right] (\theta_{k+1} - \theta_k) \end{aligned} \tag{5}$$

From the equation above, we can see that the regularization term is minimized when $\Delta$ spans the direction of low eigenvalues of the matrix $\sum_{x_j} \left[ \nabla f|_{\theta_k, x_j} (\nabla f|_{\theta_k, x_j})^T \right]$. Note that the updates on the current task can only change $\Delta$. This is similar to methods like EWC that restricts movement in direction of high curvature according to the Fisher Information matrix on previous tasks. In particular the Fisher metric considered that takes the same form as a an expectation over observations $x_j$ of the outer product of gradients. In particular we can see that this form can be interpreted (see for example (Pascanu & Bengio, 2013)) as the expected KL loss, if we consider for every $x_j$ isotropic Gaussians centered around $f(\theta_k, x_j)$ and $f(\theta_{k+1}, j)$ respectively. Note however that this expected KL is not the same as the KL between the distributions $p_X^{(k)}$ and $p_X$.

### E.2. Relationship between functional regularization and generative replay

Functional regularization (FR) and Generative replay (GR) look very similar at the first glance. For both, we take a sample from the latent space and pass it reverse through the old flow. The difference is in FR, instead of replay, we penalize the Euclidean distance between old and new embedding. In this subsection we characterize what this subtle but canonical difference may imply. Let's start with the KL distance between the old and new distribution which is indeed the loss that GR

enforces and relate it to the FR penalty term.

$$\mathcal{L}_{GR} = KL[p^{(k)}||p] = \int p^{(k)}(x|y,t) \log \frac{p^{(k)}(x|y,t)}{p(x|y,t)} dx \tag{6}$$

$$\approx \frac{1}{J} \sum_{i=1}^{J} \log \frac{p^{(k)}(x_i|y,t)}{p(x_i|y,t)} \tag{7}$$

$$= \frac{1}{J} \sum_{i=1}^{J} \left( \log \frac{p_Z(f_{\theta^{(k)}}(x_i)|y,t)}{p_Z(f_\theta(x_i)|y,t)} + \underbrace{\log \frac{\left|\frac{\partial f_{\theta^{(k)}}}{\partial x}\right|}{\left|\frac{\partial f_\theta}{\partial x}\right|}}_{\approx 0} \right) \tag{8}$$

$$\approx \frac{1}{J} \sum_{i=1}^{J} \left( \log \frac{p_Z(f_{\theta^{(k)}}(f_{\theta^k}^{-1}(z_i))|y,t)}{p_Z(f_\theta(f_{\theta^k}^{-1}(z_i))|y,t)} \right) \tag{9}$$

$$= \frac{1}{J} \sum_{i=1}^{J} \left( \log \frac{p_Z(z_i|y,t)}{p_Z(f_\theta(f_{\theta^k}^{-1}(z_i))|y,t)} \right) \tag{10}$$

$$= \frac{1}{J} \sum_{i=1}^{J} \left( \log p_Z(z_i|y,t) - \log p_Z(f_\theta(f_{\theta^k}^{-1}(z_i))|y,t) \right) \tag{11}$$

$$= \frac{1}{J \cdot \sigma^2} \sum_{i=1}^{J} \left( \|f_\theta(f_{\theta^k}^{-1}(z_i)) - \mu_y^t\|^2 - \|z_i - \mu_y^t\|^2 \right) \tag{12}$$

$$\leq \frac{1}{J \cdot \sigma^2} \sum_{i=1}^{J} \|f_\theta(f_{\theta^k}^{-1}(z_i)) - z_i\|^2 \tag{13}$$

$$= \mathcal{L}_{FR} \tag{14}$$

Note that in Eq. (8) the log of the determinant ratio is approximately assumed to be zero since the replay implicitly discourages changes in the flow mapping. The above derivation indicates a few key points on the relation between FR and GR:

- The FR loss provides an upper bound on the GR loss and is thus more restrictive; low FR loss implies low GR loss but the reverse does not necessarily hold.

- Comparing the approximation of GR in Eq. (12) and FR in Eq. (13) indicates that in FR we are pushing latent representations to the same point as they were before, while in GR the relative relocation with respect to the Gaussian centers are being pushed to be similar. In other words, GR is roughly a mean-normalized FR.

- FR does not rescale the loss according to the log determinant. That is, it doesn't take into account how the flow contracts or expands. In contrast, this stretch is being well captured by GR loss.

- In return, FR experiences a more stable loss function for the sake of optimization and convergence specially when the determinant is close to 0.

- Similarly, the sample estimate of the gradient exhibits less variance in FR compared to GR which is of practical value.

To further make sense of the relationship between FR and GR we visualize their associated loss function on a toy example, where we are mapping from univariate Gaussian (with variance 1) to another univariate Gaussian. In particular, we can afford to parametrize the flow as

$$f(\theta, x) = \theta x, \theta \in \mathcal{R}^+,$$

where $\theta > 0$ is a positive real number. In particular, let's assume we are regularizing to a previous version of the flow with parameter $\theta^{(k)} = \gamma$. In this case the loss for the function regularization quickly degenerates to

$$\mathcal{L}_{FR} = \sum_z ||f(\theta, f^{-1}(\gamma, z) - z||^2 = \sum_z ||\frac{\theta}{\gamma}z - z||^2 \approx (\theta - \gamma)^2$$

In contrast the loss for generative replay will have the form

$$\mathcal{L}_{GR} \approx \frac{1}{2}||\frac{\theta}{\gamma}z||^2 - \log(|\theta|)$$

Figure 5 shows these two losses as a function of $\theta$. Note the degeneracy of GR loss around $\theta = 0$ and how the FR loss becomes more tractable and well behaved by compromising the flow contraction/expansion from consideration.

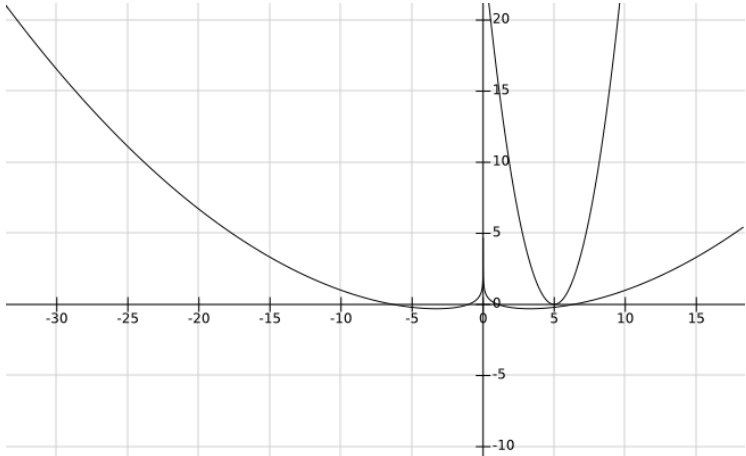

*Figure 5.* GR loss vs. FR loss

## F. Related Work

**Continual Learning**   Following Lange et al. (2019), we review the related methods to alleviate catastrophic forgetting in continual learning in three different but overlapping categories. *Replay*-based methods store and rephrase a memory of the examples or knowledge learned so far (Rebuffi et al., 2016; Lopez-Paz & Ranzato, 2017; Shin et al., 2017; Riemer et al., 2018; Rios & Itti, 2018; Zhang et al., 2019; Chaudhry et al., 2020; Balaji et al., 2020). *Regularization*-based methods constrain the parameter updates while learning new tasks to preserve previous knowledge. They include many popular and new methods such as EWC (Kirkpatrick et al., 2017), function-space regularization (Titsias et al., 2019); feature regularization, and feature replay methods (Pomponi et al., 2020a; van de Ven et al., 2020; Pomponi et al., 2020b); and orthogonality-based regularized replay methods such as OGD (Farajtabar et al., 2020), AGEM (Chaudhry et al., 2018) and GPM (Saha et al., 2021). A few works also look at continual learning from the perspectives of the loss landscape (Yin et al., 2020) and dynamics of optimization (Mirzadeh et al., 2020; Mirzadeh et al., 2020b). *Modularity-based* methods allocate different subsets of the parameters to each task (Rusu et al., 2016; Yoon et al., 2018; Jerfel et al., 2019; Li et al., 2019; Wortsman et al., 2020; Mirzadeh et al., 2020a).

**Task-Agnostic CL**   Recently, several methods have been developed for the task-agnostic CL setting. Zeno et al. (2018) and He et al. (2019) use the online variational Bayes framework to avoid the need or explicit task identities. Aljundi et al. (2019), an early advocate of task-free CL, detect task changes as peaks in the loss values following a plateau. Jerfel et al. (2019) infer the latent tasks within a Dirichlet process mixture model. Ye & Bors (2020) embed the information associated with different domains into several clusters. Mundt et al. (2019) propose a method based on variational Bayesian inference that combines a joint probabilistic encoder with a generative model and a linear classifier to distinguish unseen unknown data from trained known tasks. Achille et al. (2018) employ a variational autoencoder with shared embeddings which detects shifts in the data distribution and allocates spare representational capacity to new knowledge, while encouraging the learned representations to be disentangled. Buzzega et al. (2020) combine reservoir sampling data replay and model distillation for training models without knowing task boundaries.

The two works most closely related to HCL are CURL (Rao et al., 2019) and CN-DPMM (Lee et al., 2020). CN-DPMM uses a Dirichlet process mixture model to detect task changes; they then use a separate modularity-based method to perform the classification. CURL uses an end-to-end framework for detecting tasks and learning on them. However, CURL is primarily developed for unsupervised representation learning and cannot be trivially extended to task-agnostic supervised continual learning; in the experiments, we show that HCL achieves superior performance to a supervised version of CURL. Both CN-DPMM and CURL use a variational auto-encoder (Kingma & Welling, 2013) to model the data distribution. HCL, on the other hand, uses a single probabilistic hybrid model based on a normalizing flow to simultaneously learn the data distribution, detect task changes and perform classification.

**Out-of-Distribution Detection**   In HCL, in the task agnostic setting we need to detect data coming from new tasks, which can be viewed as out-of-distribution (OOD) detection (see e.g. Bulusu et al., 2020, for a recent survey). In particular, HCL detects task changes by measuring the typicality of the model's statistics, which is similar to recently proposed state-of-the-art OOD detection methods by Nalisnick et al. (2019c) and Morningstar et al. (2020). In some of our experiments, we apply HCL to embeddings extracted by a deep neural network; Zhang et al. (2020) develop a related method for OOD detection, where a flow-based generative model approximates the density of intermediate representations of the data. Kirichenko et al. (2020) also show that normalizing flows can detect OOD image data more successfully if applied to embeddings.

**Hybrid Models**   HCL is a hybrid generative-discriminative model that simultaneously learns to generate realistic samples of the data and solve the discriminative classification problem. Architecturally, HCL is most closely related to the semi-supervised flow-based models of Izmailov et al. (2020) and Atanov et al. (2019). These works do not consider continual learning, and focus on a very different problem setting. Nalisnick et al. (2019b) and Kingma et al. (2014) provide another two examples of hybrid models for semi-supervised learning. Zhang et al. (2020) develop a hybrid model for OOD detection.

**Normalizing flows**   Normalizing flows are flexible deep generative models with tractable likelihood based on invertible neural networks. Flows model the data distribution $p_X$ as a transformation $\hat{p}_X = f_\theta^{-1}(\hat{p}_Z)$, where $\hat{p}_Z$ is a fixed density in the latent space (typically a Gaussian), and $f_\theta : \mathcal{X} \to \mathcal{Z}$ is an invertible neural network with parameters $\theta$ that maps input space $\mathcal{X}$ to the latent space $\mathcal{Z}$ of the same dimension. We can then compute the density $\hat{p}_X$ exactly using the change of variable formula: $\hat{p}_X(x) = \hat{p}_Z(f_\theta(x)) \cdot \left| \frac{\partial f_\theta}{\partial x} \right|$, where $\left| \frac{\partial f_\theta}{\partial x} \right|$ is the determinant of the Jacobian of $f_\theta$ at $x$. The architecture of the flow networks is designed to ensure cheap computation of the inverse $f^{-1}$ and the Jacobian $\left| \frac{\partial f_\theta}{\partial x} \right|$. In HCL, we use RealNVP (Dinh et al., 2017) and Glow (Kingma & Dhariwal, 2018) flow architectures due to their simplicity and strong performance. Other flow architectures include invertible Residual Networks (Behrmann et al., 2019), residual flows (Chen et al., 2019), FFJORD (Grathwohl et al., 2018), invertible CNNs (Finzi et al., 2019) and others. For a more detailed discussion of normalizing flows, please see the recent survey by Papamakarios et al. (2019).

Normalizing flows have a number of key advantages over other deep generative models that are essential for HCL. First, unlike Generative Adversarial Networks (GANs) (Goodfellow et al., 2014), flows provide a tractable likelihood that can be used for task identification together with other model statistics (Section 3.2). Second, likelihood-based models can be used for both generation and classification, unlike GANs. Moreover, flows can produce samples of higher fidelity than Variational Autoencoders (VAEs) (Kingma & Welling, 2013) and much faster than auto-regressive models (Oord et al., 2016), which is important for alleviating catastrophic forgetting (Section 3.3). Further we demonstrate that our proposed HCL outperforms CURL (Rao et al., 2019), a VAE-based CL approach.

## G. Task identification

**Can flows detect task changes?**   Nalisnick et al. (2019a) show that deep generative models sometimes fail to detect out-of-distribution data using likelihood, e.g. when trained on FashionMNIST dataset, both normalizing flows and VAEs assign higher likelihood to out-of-distribution MNIST data. However, they consider unsupervised OOD detection, while in our case there is label information available and for each task HCL is modeling the class-conditional distribution $p_t(x|y)$. Intuitively, the model will not be able to classify unknown task samples correctly when the data distribution shifts, so the task-conditional likelihood $\hat{p}(B|t) = \hat{p}(y|x,t)\hat{p}(x|t)$ of the batch $B$ which comes from a new task $t'$ should be low. Moreover, motivated by recent advances in OOD detection with generative models (Nalisnick et al., 2019c; Morningstar et al., 2020), we propose to detect task changes using two-sided test on HCL's multiple statistics and demonstrate that HCL is able to correctly identify task change not only in standard CL benchmarks, but also in FashionMNIST-MNIST continual learning problem, which is a more challenging scenario as identified in Nalisnick et al. (2019a). Note that prior works in continual learning which are based on a VAE model (Rao et al. (2019) and Lee et al. (2020)) rely on VAE's likelihood to

determine task change points which may not be reliable in challenging settings (Nalisnick et al., 2019a).

The proposed task detection based on measuring typicality of model's statistics demonstrated strong performance in all benchmarks experiments, detecting all existing task changes. In some cases (one of the 3 runs of HCL-FR on CIFAR-10 and CIFAR-100, and HCL-GR on split MNIST) the model identified an extra task change which did not actually happen. In these cases, the model uses multiple clusters in the latent space for modeling the same class in the same task. In practice, it did not significantly hurt the final accuracy. For the runs where spurious task changes were detected, we adjusted the computation of the overall accuracy metric by accordingly re-labelling the tasks identities. For example, if during training on $T_1$ the model identifies an extra task change and then identifies the real task change to $T_2$, we consider all clusters added during training on $T_1$ to belong to task 1 and the clusters added at identified task change to $T_2$ to belong to task 2.

**Robustness** In addition to standard CL benchmark tasks, we test HCL on FashionMNIST-MNIST and MNIST-FashionMNIST domain-incremental learning classification. Although this dataset pair was identified as a failure mode for OOD detection by Nalisnick et al. (2019a), HCL's task detection correctly identified task changes in all runs.

**Task recurrence** Next, we test the ability of HCL to not only detect the task boundaries but also infer the task identities in the presence of recurring tasks. In particular, we consider the Split CIFAR-10 embeddings dataset with the following sequence of tasks: $[T_1, T_2, T_3, T_1, T_4]$ where $T_i$ is the binary classification task between the original classes $2i - 2$ and $2i - 1$. The task $T_1$ appears twice in the sequence, and the model has to identify it as an existing rather than a new task. Both HCL-FR and HCL-GR were able to successfully identify the recurring task, achieving $95.06 \pm 0.25\%$ and $91.27 \pm 0.82\%$ final average accuracy respectively.

Table 3. Results of the experiments on **split CIFAR-10** embeddings dataset extracted using EfficientNet model pretrained on ImageNet. The dataset with 10 classes is split into 5 binary classification tasks. The methods used are MTL (multitask learning) setting, Adam (regular training without alleviating forgetting), ER (standard data buffer replay with the capacity of 1000 samples per task), CURL (Rao et al., 2019), HCL-GR (generative replay), HCL-FR (functional regularization), as well as task-agnostic versions of HCL-FR and HCL-GR.

| 15 EPOCHS PER TASK | | | | | | | | |
|---|---|---|---|---|---|---|---|---|
| TASK # | 1 | 2 | 3 | 4 | 5 | ACC AVG | FORGET AVG | FULL ACC |
| MTL | 98.87 ±0.08 | 95.90 ±0.29 | 97.48 ±0.12 | 97.40 ±0.25 | 98.82 ±0.13 | 97.69 ±0.05 | 0.75 ±0.14 | 93.61 ±0.14 |
| ADAM | 90.73 ±1.05 | 58.97 ±4.88 | 54.90 ±3.82 | 81.70 ±3.76 | 99.25 ±0.04 | 77.11 ±1.33 | 27.38 ±1.63 | 19.85 ±0.01 |
| ER | 95.75 ±0.43 | 90.60 ±1.06 | 94.62 ±0.28 | 98.57 ±0.15 | 99.20 ±0.11 | 95.75 ±0.35 | 3.78 ±0.46 | 88.27 ±0.52 |
| CURL | 88.59 ±3.85 | 73.48 ±6.00 | 84.46 ±2.82 | 95.98 ±0.64 | 97.65 ±0.41 | 88.03 ±2.15 | 12.05 ±2.79 | – |
| HCL-FR | 96.95 ±0.44 | 93.22 ±0.59 | 94.58 ±0.19 | 98.50 ±0.08 | 98.97 ±0.19 | **96.44** ±0.05 | **2.47** ±0.09 | **90.12** ±0.35 |
| HCL-GR | 93.98 ±0.27 | 85.43 ±0.25 | 93.28 ±0.92 | 98.63 ±0.16 | 99.20 ±0.08 | 94.11 ±0.21 | 5.86 ±0.24 | 80.10 ±1.21 |
| HCL-FR (TA) | 96.63 ±0.33 | 92.18 ±1.33 | 94.70 ±0.19 | 98.73 ±0.25 | 98.98 ±0.10 | 96.25 ±0.17 | 2.86 ±0.28 | 89.44 ±0.80 |
| HCL-GR (TA) | 95.47 ±0.73 | 84.88 ±1.08 | 92.40 ±0.16 | 98.32 ±0.22 | 99.23 ±0.05 | 94.06 ±0.25 | 6.05 ±0.35 | 80.29 ±0.81 |

| SINGLE-PASS (1 EPOCH PER TASK) | | | | | | | | |
|---|---|---|---|---|---|---|---|---|
| TASK # | 1 | 2 | 3 | 4 | 5 | ACC AVG | FORGET AVG | FULL ACC |
| MTL | 98.92 ±0.10 | 96.67 ±0.17 | 97.25 ±0.00 | 97.20 ±0.23 | 98.18 ±0.26 | 97.64 ±0.05 | −0.05 ±0.13 | 93.69 ±0.09 |
| ADAM | 92.22 ±1.20 | 53.35 ±0.53 | 62.53 ±3.52 | 78.92 ±6.34 | 99.13 ±0.15 | 77.23 ±1.27 | 26.87 ±1.54 | 19.83 ±0.03 |
| ER | 98.32 ±0.12 | 94.08 ±0.71 | 97.12 ±0.14 | 97.73 ±0.08 | 98.55 ±0.11 | 97.16 ±0.09 | 1.32 ±0.17 | 91.85 ±0.03 |
| CURL | 95.54 ±1.16 | 80.97 ±4.83 | 80.55 ±7.16 | 94.61 ±1.99 | 96.25 ±0.41 | 89.58 ±0.92 | 8.25 ±1.10 | – |
| HCL-FR | 95.27 ±0.46 | 88.03 ±0.40 | 93.35 ±0.73 | 98.28 ±0.31 | 98.92 ±0.06 | 94.77 ±0.09 | 4.68 ±0.13 | 85.94 ±0.01 |
| HCL-GR | 93.68 ±0.55 | 85.82 ±0.27 | 93.28 ±0.41 | 98.52 ±0.17 | 99.10 ±0.04 | 94.08 ±0.08 | 5.73 ±0.06 | 82.85 ±0.40 |
| HCL-FR (TA) | 95.35 ±0.16 | 87.12 ±0.81 | 93.40 ±0.32 | 98.27 ±0.16 | 98.90 ±0.08 | 94.61 ±0.24 | 4.85 ±0.32 | 85.72 ±0.37 |
| HCL-GR (TA) | 93.37 ±0.27 | 82.28 ±1.94 | 92.33 ±0.65 | 98.23 ±0.17 | 99.17 ±0.18 | 93.08 ±0.54 | 7.05 ±0.56 | 79.93 ±0.84 |

*Table 4.* Results of the experiments on **split CIFAR-100** embeddings dataset extracted using EfficientNet model pretrained on ImageNet. The dataset with 100 classes is split into ten 10-way classification tasks. The methods used are MTL (multitask learning) setting, Adam (regular training without alleviating forgetting), ER (standard data buffer replay with the capacity of 1000 samples per task), CURL (Rao et al., 2019), HCL-GR (generative replay), HCL-FR (functional regularization), as well as task-agnostic versions of HCL-FR and HCL-GR.

### 15 EPOCHS PER TASK

| TASK # | 1 | 2 | 3 | 4 | 5 | 6 | 7 | 8 | 9 | 10 | ACC AVG | FORGET AVG | FULL ACC |
|---|---|---|---|---|---|---|---|---|---|---|---|---|---|
| MTL | 78.77 ±0.39 | 73.30 ±1.27 | 76.53 ±1.30 | 72.33 ±0.25 | 76.87 ±0.78 | 73.63 ±2.36 | 77.63 ±1.60 | 78.73 ±0.12 | 83.53 ±0.80 | 71.87 ±0.97 | 76.32 ±0.52 | 9.37 ±0.34 | 74.11 ±0.55 |
| ADAM | 7.77 ±0.25 | 1.00 ±0.29 | 4.63 ±0.60 | 3.80 ±1.26 | 3.33 ±1.15 | 10.83 ±0.46 | 17.60 ±1.34 | 5.07 ±0.05 | 17.10 ±0.80 | 96.83 ±0.25 | 16.80 ±0.27 | 86.49 ±0.33 | 9.84 ±0.04 |
| ER | 65.70 ±1.23 | 63.57 ±1.60 | 68.83 ±0.78 | 62.97 ±1.32 | 71.93 ±0.76 | 71.73 ±0.24 | 75.20 ±0.71 | 76.40 ±1.08 | 86.20 ±0.57 | 69.37 ±1.72 | 71.19 ±0.22 | 17.74 ±0.32 | 68.46 ±0.27 |
| CURL | 11.62 ±2.31 | 3.34 ±1.07 | 4.34 ±2.01 | 4.26 ±2.02 | 6.72 ±3.23 | 18.76 ±1.25 | 22.28 ±2.41 | 31.66 ±2.02 | 55.44 ±1.19 | 82.68 ±0.52 | 24.11 ±0.72 | 65.24 ±0.68 | – |
| HCL-FR | 54.27 ±1.68 | 51.00 ±1.87 | 59.10 ±1.85 | 50.30 ±1.82 | 56.10 ±0.29 | 57.10 ±1.08 | 67.67 ±2.05 | 68.33 ±0.87 | 81.20 ±0.75 | 93.47 ±0.21 | 63.85 ±0.80 | 31.89 ±0.99 | 60.58 ±0.78 |
| HCL-GR | 44.53 ±1.18 | 41.43 ±1.75 | 56.57 ±0.25 | 47.53 ±0.59 | 52.00 ±1.77 | 53.87 ±1.24 | 68.93 ±1.60 | 68.10 ±0.43 | 82.93 ±0.24 | 94.67 ±0.19 | 61.06 ±0.43 | 36.10 ±0.56 | 57.39 ±0.60 |
| HCL-FR (TA) | 55.03 ±0.57 | 51.13 ±1.89 | 55.33 ±4.23 | 48.17 ±0.63 | 56.50 ±1.56 | 56.53 ±1.02 | 67.87 ±0.66 | 65.97 ±2.59 | 80.17 ±1.36 | 93.93 ±0.33 | 63.06 ±0.64 | 32.52 ±0.88 | 59.66 ±0.70 |
| HCL-GR (TA) | 42.10 ±0.75 | 35.17 ±2.32 | 50.37 ±0.86 | 40.77 ±0.88 | 46.13 ±0.63 | 45.83 ±1.01 | 61.87 ±0.87 | 59.33 ±0.48 | 78.47 ±0.68 | 95.30 ±0.22 | 55.53 ±0.23 | 42.46 ±0.25 | 51.64 ±0.14 |

### SINGLE-PASS (1 EPOCH PER TASK)

| TASK # | 1 | 2 | 3 | 4 | 5 | 6 | 7 | 8 | 9 | 10 | ACC AVG | FORGET AVG | FULL ACC |
|---|---|---|---|---|---|---|---|---|---|---|---|---|---|
| MTL | 86.23 ±0.19 | 81.43 ±0.54 | 83.77 ±0.73 | 79.80 ±0.36 | 78.53 ±0.37 | 72.87 ±1.64 | 73.47 ±0.54 | 63.13 ±1.54 | 60.80 ±0.70 | 35.93 ±2.52 | 71.60 ±0.38 | −12.18 ±0.13 | 68.80 ±0.52 |
| ADAM | 8.47 ±1.15 | 1.13 ±0.42 | 5.03 ±0.37 | 2.30 ±0.65 | 2.90 ±0.45 | 9.83 ±0.66 | 19.07 ±0.52 | 12.00 ±1.28 | 22.33 ±2.18 | 95.87 ±0.17 | 17.89 ±0.18 | 83.64 ±0.28 | 11.29 ±0.31 |
| ER | 78.23 ±0.79 | 76.77 ±1.22 | 80.40 ±0.42 | 80.10 ±0.45 | 80.63 ±0.26 | 74.03 ±1.17 | 72.07 ±0.12 | 62.87 ±1.22 | 58.27 ±1.70 | 17.97 ±0.76 | 68.13 ±0.36 | −27.10 ±0.25 | 65.13 ±0.25 |
| CURL | 12.82 ±4.73 | 4.94 ±1.73 | 9.88 ±2.95 | 9.38 ±2.89 | 11.52 ±2.67 | 14.44 ±4.05 | 22.44 ±3.18 | 15.64 ±1.87 | 24.38 ±6.32 | 59.98 ±9.29 | 18.54 ±1.54 | 44.00 ±2.66 | – |
| HCL-FR | 50.30 ±1.16 | 13.80 ±2.27 | 48.37 ±1.05 | 38.63 ±0.57 | 37.90 ±2.14 | 39.77 ±0.25 | 51.47 ±5.70 | 53.33 ±2.26 | 73.77 ±0.31 | 94.03 ±0.76 | 50.14 ±0.41 | 43.31 ±0.17 | 45.76 ±0.34 |
| HCL-GR | 56.10 ±1.22 | 42.67 ±0.94 | 58.93 ±1.84 | 39.63 ±1.68 | 41.53 ±2.49 | 36.13 ±1.60 | 45.83 ±4.90 | 46.00 ±2.25 | 51.40 ±6.68 | 90.87 ±1.40 | 50.91 ±0.82 | 40.40 ±0.85 | 46.10 ±0.99 |
| HCL-FR (TA) | 49.50 ±2.40 | 16.27 ±2.23 | 48.73 ±2.19 | 37.73 ±1.14 | 38.87 ±1.90 | 37.80 ±3.19 | 51.77 ±1.65 | 50.53 ±2.09 | 75.07 ±1.61 | 93.63 ±0.25 | 49.99 ±0.86 | 43.04 ±0.58 | 45.64 ±0.97 |
| HCL-GR (TA) | 35.20 ±0.91 | 30.20 ±1.99 | 41.17 ±1.92 | 27.30 ±5.62 | 35.57 ±1.14 | 33.13 ±2.83 | 43.70 ±0.43 | 45.23 ±1.27 | 68.03 ±1.00 | 95.50 ±0.43 | 45.50 ±0.46 | 52.08 ±0.57 | 40.87 ±0.57 |