# OpenReview forum: "Task-agnostic Continual Learning with Hybrid Probabilistic Models"
_ICML.cc/2021/Workshop/INNF — INNF+ 2021 spotlighttalk_

### Official Review · Reviewer_6qH5 · 2021-06-12

**Rating:** Accept
**Confidence:** 3

**Summary:**

This paper proposes HCL for continual learning using normalizing flows, in the task-agnostic scenario. It applies normalizing flow to model the data distribution. For the task-agnostic setting,  the authors propose to identify the task boundary by measuring the typicality of the HCL model's statistics. It also proposes 2 types of approaches to alleviating catastrophic forgetting.

**Justification For Rating:**

Pros:
This paper yield moderate and sufficient contribution to continual learning by introducing normalizing flow to the game. The idea is natural and also has some theoretical insights.
The empirical validations imply the usefulness of the methods compared to task-aware methods.

Suggestions:
It should be more clearly stated the difference between the proposal and prior work. For instance, it's not that clear to me from the main text whether the paper proposed the initial idea of applying normalizing flow to continual learning. If it is, it's better to emphasize that.

---

### Official Review · Reviewer_oiQN · 2021-06-12

**Rating:** Accept
**Confidence:** 3

**Summary:**

The paper proposes HCL, a Hybrid generative-discriminative approach to Continual Learning for classification. By leveraging the properties of normalizing flows, it models each task as a gaussian in the hidden space. Once a new task appears, it initializes a new gaussian for the task.

**Justification For Rating:**

The idea of the paper is novel and interesting. It may be important to the community. The paper is easy to read. The pictures illustrate the method, it helps a lot. The experiments justify the statements of the authors.

---

### Official Review · Program_Chairs · 2021-06-15

**Rating:** Borderline Accept
**Confidence:** 3

**Summary:**

This work proposes a framework (model + collection of losses) for continual supervised learning of multiple tasks. It proposes to use a flow-based density model of a joint density p(x,y,t) between inputs X, labels Y and tasks T. Additionally, it uses the likelihood statistics to detect a task change.

**Justification For Rating:**

The paper is a bit off-topic for the workshop, as it does not focus on the flow itself but on a bigger framework in which the flow is just one ingredient.

The text is overall well organised and clear. The problem of continual learning and task change detection is relevant for ML in general.

The experiments are well done and well explained. They show good results but not substantially better than the CURL baseline.

The model training setup is well-motivated, although the combination of various losses is a bit ad-hoc.

---

### Decision · Program_Chairs · 2021-06-14

Accept (spotlight talk)